# Analysis of Pre-Seismic Ionospheric Disturbances Prior to 2020 Croatian Earthquakes

**Mohammed Y. Boudjada [1,\*], Pier F. Biagi [2], Hans U. Eichelberger [1], Giovanni Nico [3], Patrick H. M. Galopeau [4], Anita Ermini [5], Maria Solovieva [6], Masashi Hayakawa [7,8], Helmut Lammer [1], Wolfgang Voller [1] and Martin Pitterle [1]**

1 Space Research Institute, Austrian Academy of Sciences, 8042 Graz, Austria;
   hans.ulrich.eichelberger@oeaw.ac.at (H.U.E.); helmut.lammer@oeaw.ac.at (H.L.);
   wolfgang.voller@oeaw.ac.at (W.V.); martin.pitterle@oeaw.ac.at (M.P.)
2 Department of Physics, University of Bari, 70126 Bari, Italy; pf.biagi@gmail.com
3 Institute for Applied Mathematics (IAC), National Research Council of Italy (CNR), 70126 Bari, Italy;
   g.nico@ba.iac.cnr.it
4 LATMOS-CNRS, UVSQ Université Paris-Saclay, 78280 Guyancourt, France; patrick.galopeau@latmos.ipsl.fr
5 Department of Industrial Engineering, University of Tor Vergata, 00133 Rome, Italy; ermini@uniroma2.it
6 Institute of the Earth Physics, Russian Academy of Sciences, Moscow 123995, Russia; rozhnoi@ifz.ru
7 Hayakawa Institute of Seismo Electromagnetics, Co., Ltd. (Hi-SEM), Tokyo 182-8585, Japan;
   hayakawa@hi-seismo-em.jp
8 Advanced Wireless & Communications Research Center, The University of Electro-Communications,
   Tokyo 182-8585, Japan
\* Correspondence: mohammed.boudjada@oeaw.ac.at

**Abstract:** We study the sub-ionospheric VLF transmitter signals recorded by the Austrian Graz station in the year 2020. Those radio signals are known to propagate in the Earth-ionosphere waveguide between the ground and lower ionosphere. The Austrian Graz facility (geographic coordinates: 15.46°E, 47.03°N) can receive such sub-ionospheric transmitter signals, particularly those propagating above earthquake (EQ) regions in the southern part of Europe. We consider in this work the transmitter amplitude variations recorded a few weeks before the occurrence of two EQs in Croatia at a distance less than 200 km from Graz VLF facility. The selected EQs happened on 22 March 2020 and 29 December 2020, with magnitudes of $M_w$5.4 and $M_w$6.4, respectively, epicenters localized close to Zagreb (16.02°E, 45.87°N; 16.21°E, 45.42°N), and with focuses of depth smaller than 10 km. In our study we emphasize the anomaly fluctuations before/after the sunrise times, sunset times, and the cross-correlation of transmitter signals. We attempt to evaluate and to estimate the latitudinal and the longitudinal expansions of the ionospheric disturbances related to the seismic preparation areas.

**Keywords:** earthquakes precursors; VLF transmitter signals; sub-ionospheric wave propagation

## 1. Introduction

Earthquake (EQ) prediction using radio techniques is recognized as an effective tool to investigate and characterize seismo–electromagnetic phenomena [1]. Radio techniques are mainly based on the study of the radio wave propagation in the Earth's waveguide, where the seismic areas are considered to be localized between the emitting transmitter and the receiving antenna. In this context, three distinctive propagations can be taken into account: the ground wave, and the tropospheric and the ionospheric propagations [2]. In this work, we consider very low frequencies (VLFs) and low frequencies (LFs), which range from 3 kHz to 300 kHz and correspond to a wavelength from 100 km to 1 km. Such frequencies have the advantage of being reflected by the ionospheric D- and E-layers with fairly slow fading rates. The earth–ionosphere waveguide (WGEI) is a key region, which plays a crucial role, particularly in rendering propagation possible over distances of a range

of 5000 km up to 20,000 km [3]. WGEI models lead to the study of ionospheric seismic disturbance prior to EQ occurrences [4].

A lot of attention has been given to VLF/LF propagation after the detection by [5] of seismo–ionosphere anomalies in the amplitude and phase of the Tsushima Omega transmitter signal (Japan), who investigated the Kobe EQ (Japan, 17 January 1995, $M_w7.2$) using Tsushima VLF Omega transmitter signals detected by the Inubo receiving station [5]. The distance transmitter–Inubo station was about 1000 km. The diurnal variations of the amplitude and phases of the transmitter exhibit minima at the terminator points, two times a day at local sunrise and sunset. The developed method called the terminator time (TT) method is defined as the time when the transmitter amplitude or the transmitter phase variations reveal a minimum, around local sunrise and sunset. This signifies that the minima of amplitudes/phases occurred before or after the local sunrise or sunset. Authors estimated the times of the sunrise and sunset terminators a few days before, and showed the presence of a remarkable shift. This result has been confirmed by ([5–7]) using VLF subionospheric signals in the case of ten EQs of magnitude bigger than 6, and they concluded that the terminator shifts correspond to a change in the ionosphere, which has been lowered by a few kilometers. Later on, the terminator time method (hereafter TTs) has been considered in a number of studies where anomalies were reported in the cases of the Sumatra $M_w9.3$ EQ in 2004 [8], the Pakistan $M_w7.4$ EQ in 2011 [9], the Honshu Japan $M_w9.0$ EQ in 2011 [10], and the Nepal $M_w7.3$ EQ in 2015 [11].

The initial explanation of the TTs shift attributes it to the change in ionospheric electron density, as previously reported. Yoshida et al. (2008) [12] studied TTs data with a wave-hop method for a relatively short propagation path, less than 2000 km. They [12] showed that the ionosphere lowering decreases the sky wave path length, which modifies the interference condition of this wave with the ground wave. Yoshida et al. (2008) [12] concluded that TTs are the consequence of the destructive interference between the sky and ground waves. Further investigations applied harmonic analysis to TT observations allowing the exhibition of the presence of enhanced modulations. These variations revealed periodicities of 5 days or 9–11 days [13], who linked those modulations to planetary waves generated by atmospheric oscillations. Such waves provided coupling between the lithosphere and the ionosphere. In the year 2004, Molchanov et al. (2004) [14] and Pulinets and Boyarchuk (2004) [15] detailed and discussed the lithosphere–atmosphere–ionosphere coupling (LAIC) model, which is considered as a leading mechanism for EQ precursors in the atmosphere and the ionosphere. The change in physical parameters in the atmosphere, such as temperature and density, could track preseismic water/gas release, resulting in the formation of atmospheric gravity waves (AGWs) with periods in an interval of 6–60 min ([16–18]). Such seismo-induced waves could allow for ionospheric turbulence followed by the modification of the over-the-horizon radio wave propagation in the atmosphere and lower ionosphere [13]. More recently, Rapaport et al. (2022) [19] retrieved quasi-wave oscillations associated to AGWs with period of 4–10 and 20–25 min, considered to be generated by EQs or solar terminators. They [19] found that the information entropy increased at sunset and sunrise with seasonal variation.

A complementary approach has been examined by [14], who considered other atmospheric elements, like aerosols, natural radioactivity, and atmospheric electricity. They insisted on the coupling linked to active tectonic faults, around the EQ epicenter, which release radon and other radioactive gases. This leads to the generation of alpha particles, formed principally by radon decay, and consequently an ionization of the lower atmosphere. Radon is assumed to be the principal source of boundary–layer modifications through the air ionization, which generates two key branches, according to the model of [20]: the thermal anomalies through the geochemical–thermal interface and the ionospheric and magnetospheric anomalies through the geochemical–electromagnetic interface. More issues, requirements, and constraints have recently been addressed in both models (i.e., generation of AGW and radon ionization of the atmosphere) by [21–23].

Additionally, new investigations using total electron content (TEC) derived from satellite observations have provided further details about the coupling between the lithosphere and the ionosphere. Statistical analysis of more than 500 EQs [24] shows that seismic events induced TEC anomalies, and for specific cases, the presence of an equatorial ionization anomaly in the latitudinal TEC profiles at longitudes surrounding the epicenter [25]. Such TEC anomalies have been interpreted as seismic variations acting in the form of emanation of energy from the lithosphere to the ionosphere [26]. TEC measurements were combined with crustal stress (the b-value) in the case of the $M_w$7.7 Colima EQ of 19 September 2022; [27] found that the regions with lower TEC values and b-values are subject to an elevated probability of significant EQs.

In this paper, we apply the terminator method to two EQs that occurred in the vicinity of Zagreb, Croatia. In the Section 2, we provide the main features of the two seismic events, a description of the INFREP European network, and the main steps followed for processing the VLF/LF transmitter signals. The main outcomes are detailed in Section 3, where the expected preseismic areas are displayed, taking into consideration the anomaly variations related to the time shift fluctuations of the transmitter signals. The main results are discussed in Section 4 and summarized in Section 5.

## 2. VLF Transmitter Signal Variations before Croatian EQ Occurrences

We introduce in this section the main seismic features related to the two EQs that occurred in Croatia on 22 March 2020 and 29 December 2020, followed by a report on the detected transmitter signals recorded by the VLF/LF reception system in Graz, Austria. The method is detailed and applied to process the selected transmitter signals (i.e., ICV, ITS, TBB, and RRO transmitters) that surround the Croatian EQ areas.

### 2.1. Features of Croatian Earthquakes on 22 March and 29 December 2020

We analyze two events that occurred in Croatia in the year 2020, as shown in Figure 1. The first earthquake, hereafter called Event-1, happened on 22 March 2020 at 05:24 UT with a magnitude of $M_w$5.4 and a depth of 10 km. The second one, hereafter called Event-2, occurred on 29 December 2020 at 11:29 UT with magnitude 6.4 and a depth similar to Event-1. The geographic coordinates of the first and second events are, respectively, 45.87°N and 45.42°N in latitudes, and 16.02°E and 16.21°E in longitudes. The distances to Zagreb (Croatia) were 7 km and 47 km for Event-1 and Event-2, respectively.

### 2.2. INFREP VLF and LF European Network

The International Network for Frontier Research on Earthquake Precursors (INFREP) was established more than 10 years ago in the Southern part of Europe, initially with six stations, i.e., two stations in Italy, two in Greece, and two in Romania [28]. Later on, three others were installed in Austria, Cyprus, and Portugal [29]. Elettronika Italian Company produced the INFREP reception system, i.e., antennas and receivers. An additional complementary reception system [30], called hereafter Ultra-MSK, has also been used in this study. Both reception systems are localized in the Graz ground facility at a latitude of 47.03°N and longitude of 15.46°E. We have used in this study the following transmitter signals, which are monitored daily by the INFREP and Ultra-MSK systems: ICV (20.27 kHz, 40.91°N and 09.71°E, Tavolara, Italy), ITS (45.9 kHz, 37.13°N and 14.44°E, Niscemi, Italy), TBB (37.4°N and 27.21°E, Bafa, Turkey), and RRO (45.75°N and 25.61°E, Eforie, Rumania).

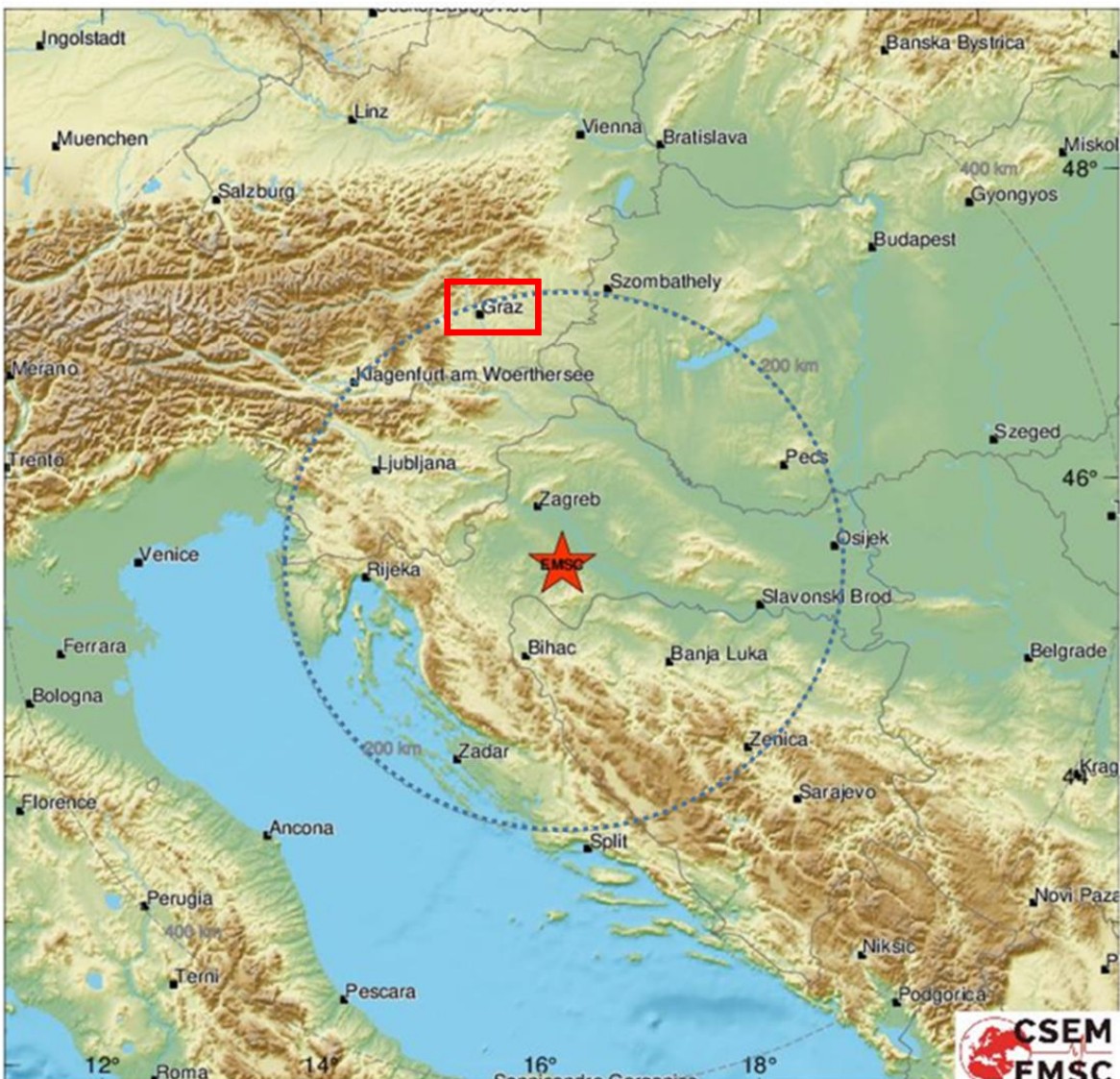

**Figure 1.** The locations of the EQ epicenters (red star) and the Graz VLF/LF facility (red square). The seismic events occurred on 22 March and 29 December 2020 at distances, respectively, of about 136 km and 186 km. The EQ surrounding environments are designated by the blue dashed circle with radii equal to 200 km.

### 2.3. Methodology Applied to VLF/LF Signal Analysis

We analyze in this study the diurnal variations of the transmitter signal amplitudes detected by INFREP and Ultra-MSK systems where the following steps are examined. The first step consists of investigating the amplitude variations at 13 and 3 days, respectively, before and after the EQ occurrence. As an example, we show in Figure 2 the daily variations of the TBB transmitter signal from 10 March 2020 to 25 March 2020 corresponding to 070 and 085 day of the year 2020 (hereafter DOY). Generally, the signal is modulated by the daily Earth's rotation where the amplitude is high, up to 43 dB, during the nighttime and decreases down to $-110$ dB in the daytime. The ionospheric D-layer is at the origin of the signal attenuation during the daytime observations. The second step is dedicated to the study of the anomalies' occurrence for both events, where the time shifts of sunrise and sunset terminators are derived from the transmitter signal variations.

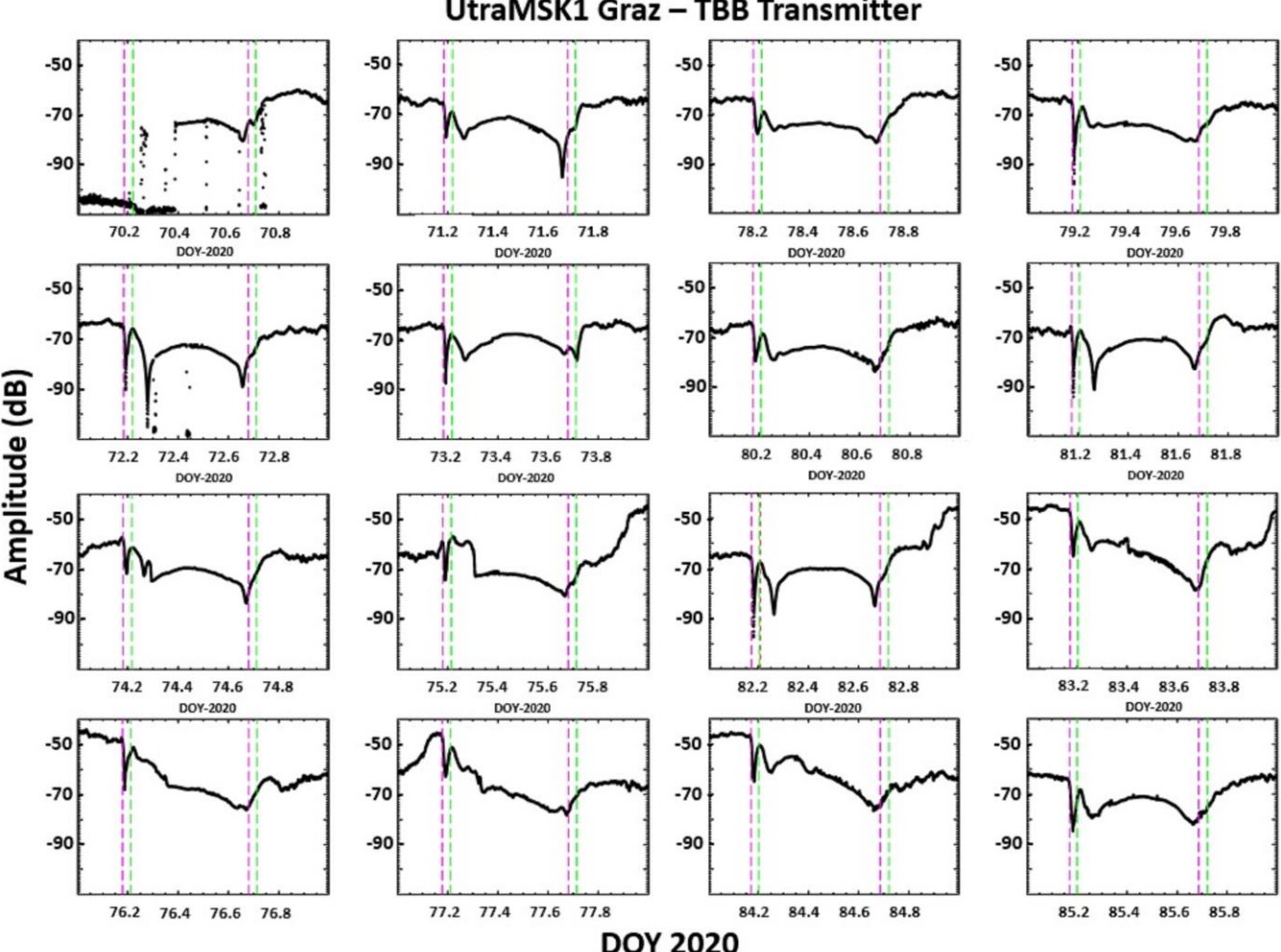

**Figure 2.** TBB transmitter signal variations recorded by the Ultra-MSK system from 10 March 2020 to 25 March 2020, which correspond, respectively, to 70 DOY and 85 DOY. The horizontal and vertical axes indicate the time (24 h) and the amplitude from −110 dB to −40 dB. The vertical lines designate the sunrises and sunsets at the TBB transmitter location (violet dashed vertical lines) and the sunrises and sunsets at the Graz VLF/LF facility (green dashed vertical lines). The EQ occurred on 22 March 2020 (82 DOY) at 05:24 UT.

The third step is devoted to the analysis of the expected locations of preseismic areas by considering that the VLF/LF investigated signals are supposed to be emitted circularly and isotropically. A cross-correlation with a lag of 16 days has been applied to both events, in the fourth step, with the aim to emphasize the relationship between the transmitter signals and the derived anomaly days.

## 3. Anomalies as Derived from Terminator Time Investigations

This section is devoted to the analysis of the anomalies that happened about 13 days before the EQs occurrence. The anomaly is defined, according to the work of Hayakawa et al. (1996) [5], as a shift of the sunrise and/or sunset terminators. This time shift is derived from the VLF amplitude minima occurring before/after sunrise and sunset terminators.

Figures 3 and 4 display the shift terminators as derived from the minima of VLF/LF transmitter signals for Event-1 and Event-2, respectively. The horizontal and vertical axes designate the observation time (expressed in DOY) and the terminator shifts (expressed in hours). The green, red, and blue curves, in each panel, refer to the transmitter terminator (e.g., ITS transmitter sunrise in the first panel in Figure 3), to the Graz terminator (e.g., sun-

rise in the first panel in Figure 3) and the shift terminator derived from the transmitter signal (e.g., ITS transmitter in first panel in Figure 3), respectively. The terminators of the Graz station and transmitters are derived from spherical trigonometry equations [31] taking into consideration their geographical coordinates and UTs.

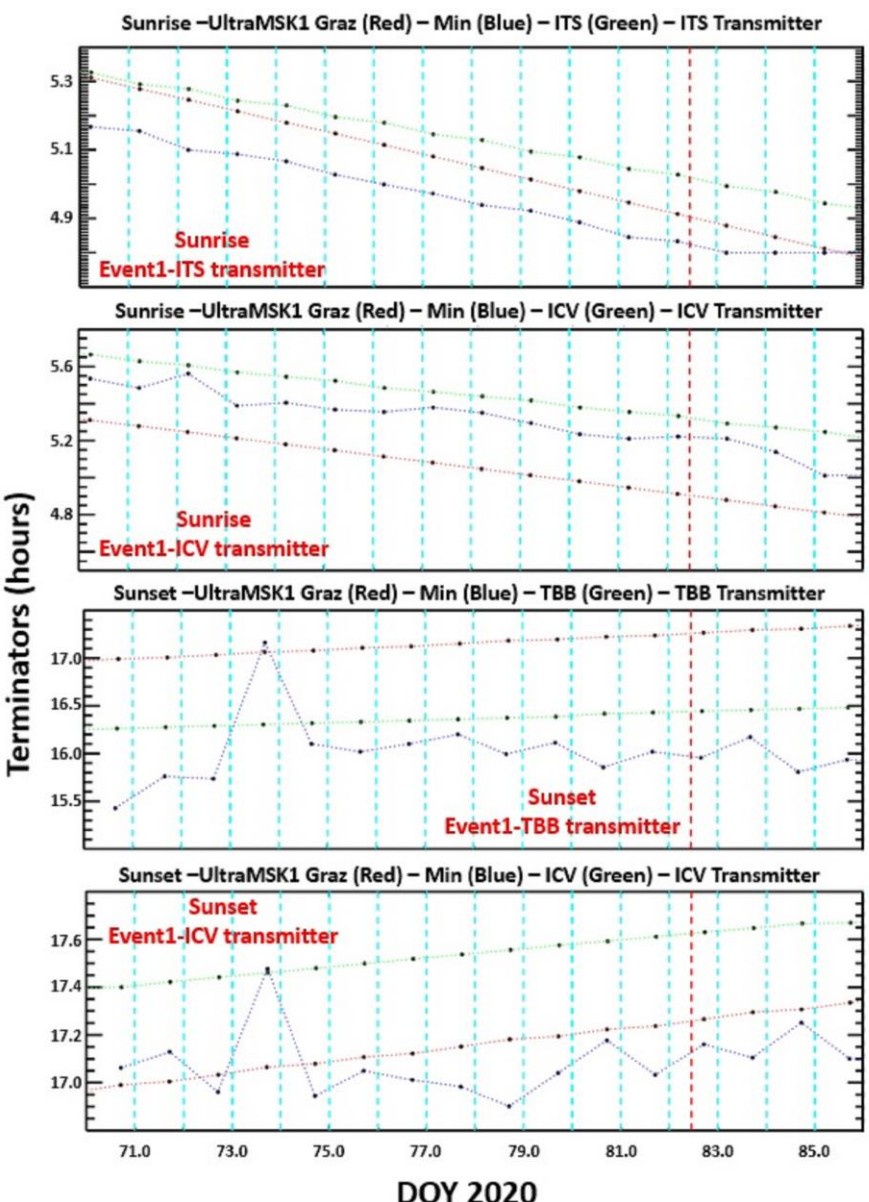

**Figure 3.** The variation in the terminator shift in the case of the Event-1 where the horizontal and the vertical axes correspond the observation time (expressed in DOY) and the terminator shifts (expressed in hours). The first and second panels show fluctuations of ITS and ICV at sunrise, and the third and fourth panels at sunset for the TBB and ICV transmitters. The red dashed vertical line indicates the day of the EQs, i.e., 22 March 2020 (082 DOY). The green, red, and blue curves, in each panel, correspond to the transmitter terminator (e.g., ITS transmitter sunrise in the first panel in this Figure), to the Graz terminator (e.g., sunrise in the first panel in this Figure) and the shift terminator derived from the transmitter signal (e.g., the ITS transmitter in first panel in this Figure), respectively.

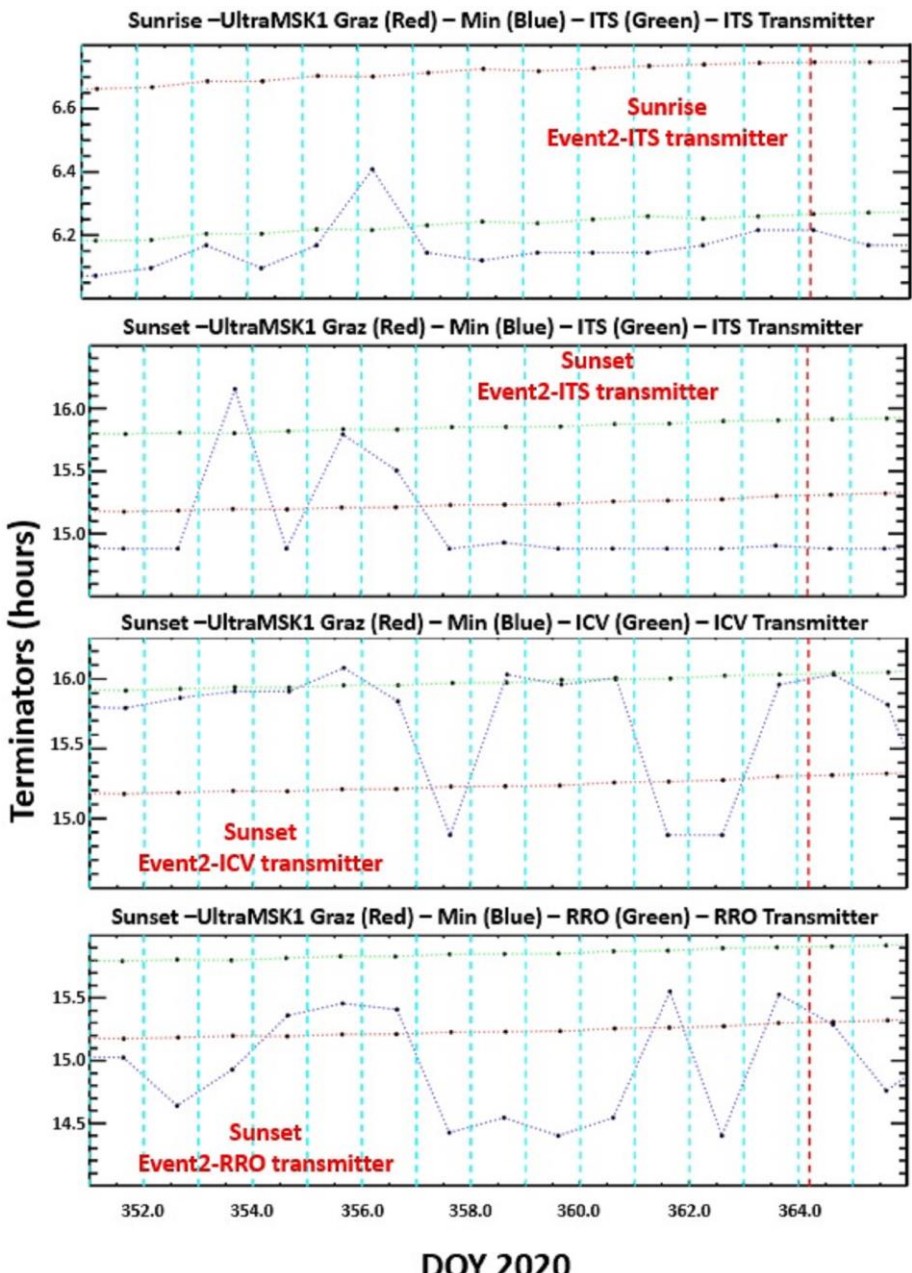

**Figure 4.** As Figure 3, but Event-2. The first panel shows the variation in the ITS transmitter at sunrise, and the other panels display sunsets for ITS, ICV, and RRO. The red dashed vertical line indicates the day of EQs, i.e., 29 March 2020 (364 DOY).

### 3.1. Anomalies Associated to the EQ of 22 March 2020

The investigated period starts on 10 March (070 DOY) and ends on 25 March (085 DOY) 2020. The first panel of Figure 3 displays the variation in ITS shift terminators at sunrise. We note that the shift (blue dashed curves) occurs earlier when compared to the Graz and ITS sunrise terminators. The shift varies from 1 min (085 DOY) to about 8 min with regard to the Graz terminator. It is different when considering ICV transmitter signal, as shown in the second panel, where the shift appears between the ICV and Graz station terminators. The time intervals are 10–19 min and 2–14 min with regard to the Graz and ICV sunrise terminators, respectively. The TBB transmitter and Graz station sunset terminators exhibit a difference of about 45 min due to their longitude locations, as displayed in the third panel of Figure 3. In this case, the anomaly appears, first, before the Graz station terminator (070 DOY–072 DOY), jumps both the Graz and TBB terminators (073DOY), and emerges again

before the Graz station terminator (074 DOY–085 DOY). Similar situation occurs for ICV and Graz station sunset terminators, where a jump happens on 073 DOY.

### 3.2. Anomalies Associated with the EQ of 29 December 2020

For the Event-2, we consider three different transmitters where the anomalies have been observed from 16 December 2020 (351 DOY) to 31 December 2020 (366 DOY). Figure 4 displays, in the first panel, the variation in the shift sunrise terminator of ITS transmitter signal. The shift appears near to the ITS terminator with a time interval 2–7 min, and far from the Graz terminator with an interval of 31–36 min. However, a jump occurs on 356 DOY, where the anomaly appears between the ITS and Graz terminators. The three further panels of Figure 4 display the shift of sunset terminators. Like in the first panel, several jumps happen on 353 DOY, 355 DOY, and 356 DOY for the ITS-transmitter anomaly (second panel), on 357 DOY, 361 DOY, and 362 DOY for the ICV-transmitter anomaly (third panel), and on 354 DOY, 355 DOY, and 356 DOY for the RRO-transmitter anomaly (fourth panel).

Table 1 summarizes the main jumps, as derived from Figures 3 and 4. The investigated event is given in the first column, the sunrise or sunset in the second column, the transmitter where the anomaly appears in the third column, and the last three columns indicate, respectively, the observation day (i.e., DOY) and the time difference (expressed in minutes), the transmitter–terminators and anomalies (i.e., shift1), and the Graz-terminator and anomalies (i.e., shift2).

**Table 1.** List of the anomalies jumps as derived from Figures 3 and 4.

| EQ Event | | Anomaly in Transmitter | DOY | Shift1 min | Shift2 min |
|---|---|---|---|---|---|
| Event-1 | Sunset | TBB | 073 | 51 | 05 |
| 22 March 2020 | Sunset | ICV | 073 | 01 | 24 |
| | | | | | |
| Event-2 | Sunrise | ITS | 356 | 11 | 18 |
| 29 December 2020 | Sunset | | 353 | 12 | 57 |
| | Sunset | ICV | 357 | 65 | 20 |
| | Sunset | | 361 | 67 | 23 |
| | Sunset | | 362 | 68 | 24 |
| | Sunset | RRO | 354 | 45 | 10 |
| | Sunset | | 355 | 50 | 14 |
| | Sunset | | 356 | 47 | 11 |
| | Sunset | | 361 | 52 | 17 |
| | Sunset | | 363 | 50 | 13 |

### 3.3. Expected Locations of Preseismic Areas

Figure 5 displays the locations of the investigated transmitter signals with regards to the Graz VLF/LF facility. In the idealistic case, the emitted VLF/LF radio transmitter wave is supposed to be radiated circularly and isotropically, where the central part is supposed to be the transmitter location. This simple sketch is used to estimate the location of the preseismic area at the origin of the observed anomalies. Figures 6 and 7 are zoomed from Figure 5. We aim here to display the preseismic geographic locations taking into consideration the shift anomalies as shown in Figures 3 and 4.

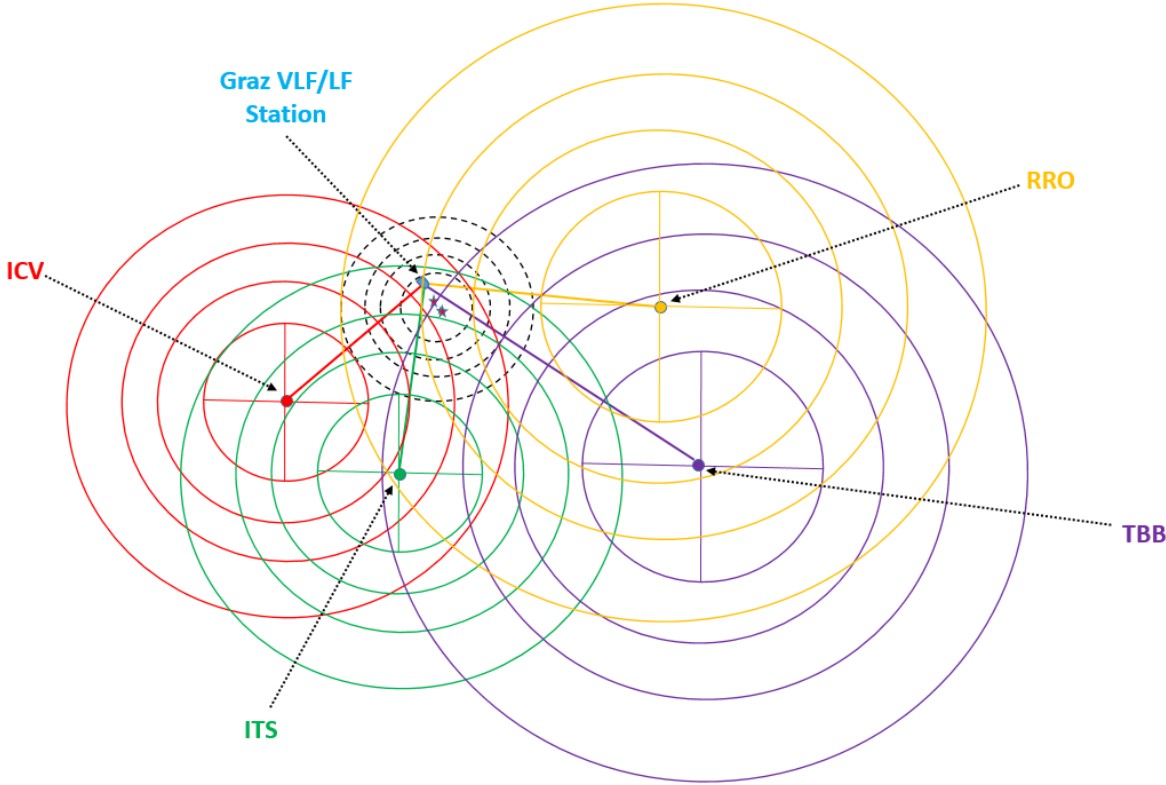

**Figure 5.** Simple sketch of the idealistic case where the investigated VLF/LF transmitters are supposed to emit radiation circularly and isotropically. The two red stars close to Graz VLF/LF facility indicate the location of the two EQs, which occurred in the year 2020 in the vicinity of Zagreb, Croatia. The full red, green, violet, and yellow colors small circles, shown with dashed black arrows, designate to the locations of ICV, ITS, TBB, and RRO transmitters, respectively.

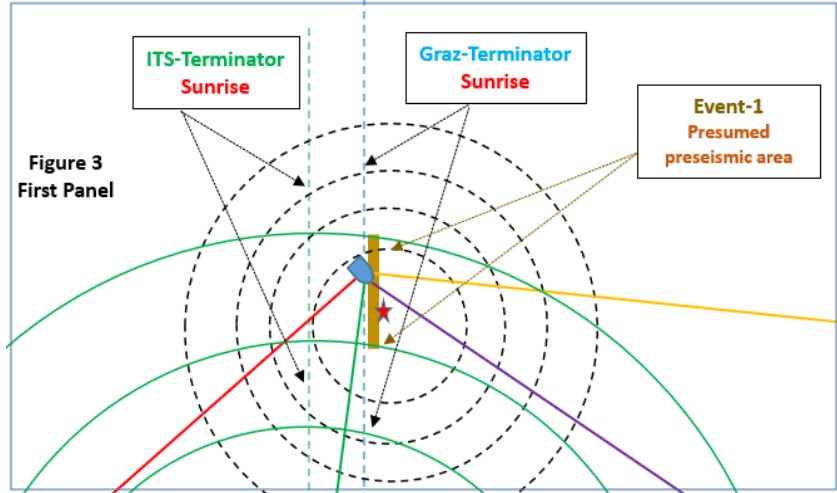

**Figure 6.** *Cont.*

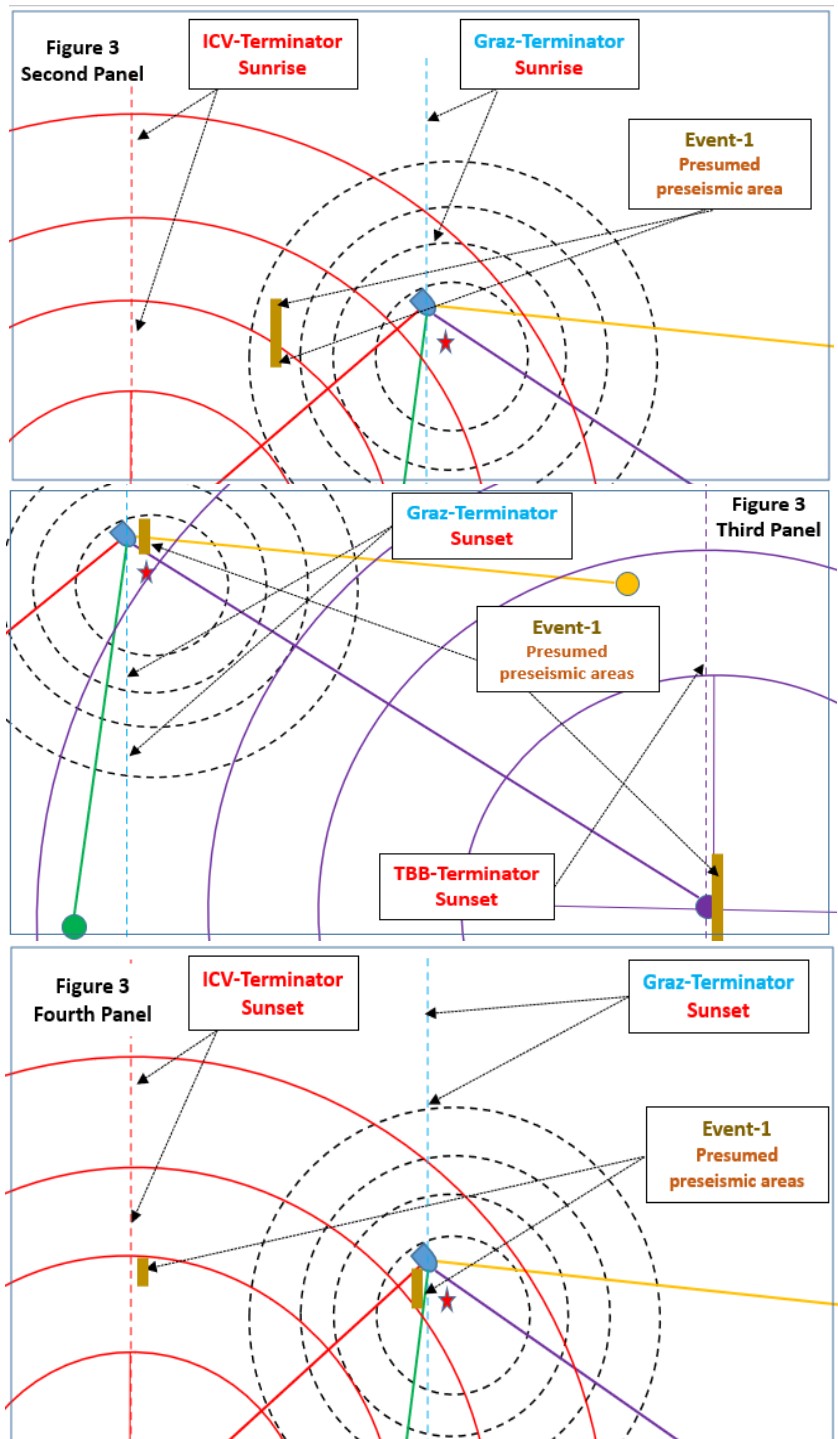

**Figure 6.** Presumed locations of the preseismic areas taking into consideration the anomalies as recorded by the Graz facility and displayed in Figure 3 in the case of the earthquake (i.e., red stars), which occurred on 22 March 2020 (i.e., Event-1). The blue, red, green, and violet vertical dashed lines indicate, respectively, the local sunrise or sunset terminators of the Graz station, ITS, ICV, and TBB transmitters.

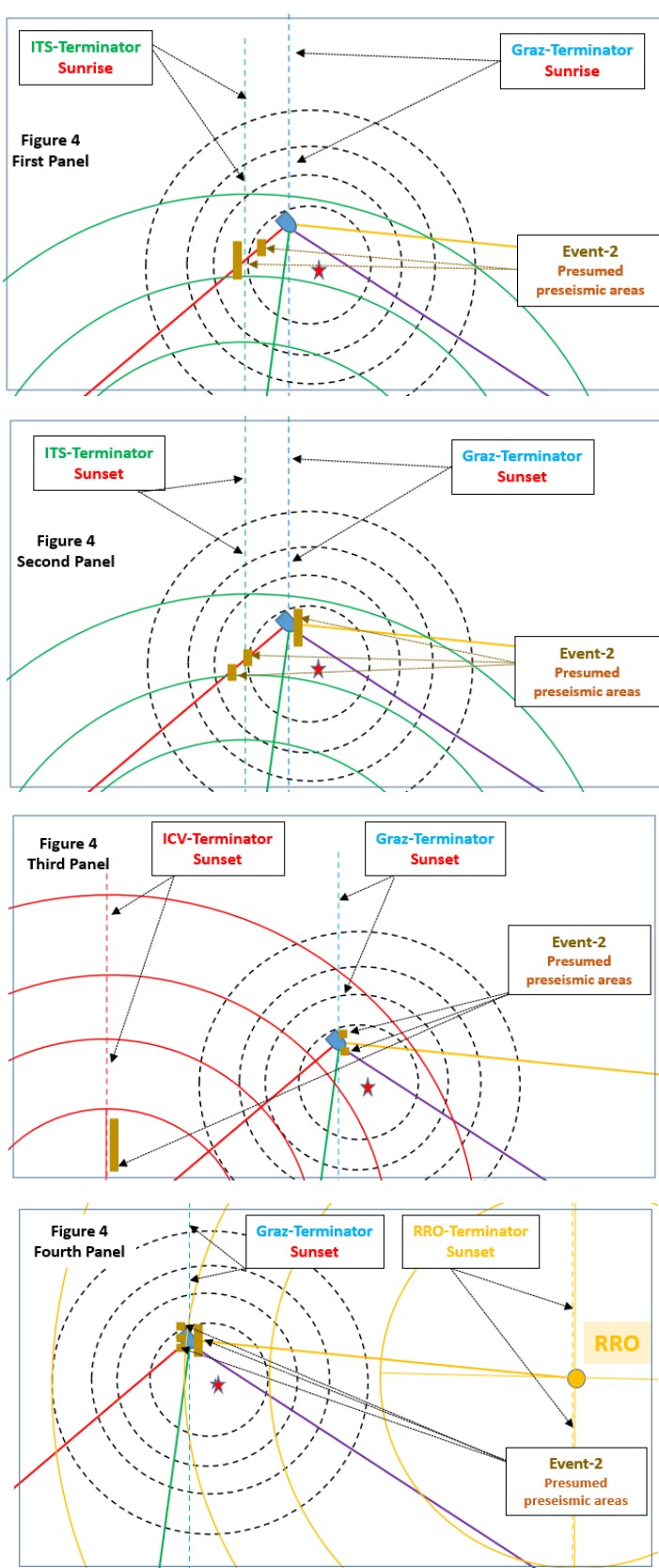

**Figure 7.** Like in Figure 6, presumed locations of the preseismic areas taking into consideration the anomalies as shown Figure 4 in the case of the EQs (i.e., red star), which occurred on 29 December 2020 (i.e., Event-2). The blue, red, green, and yellow vertical dashed lines indicate, respectively, the local sunrise or sunset terminators of the Graz station, ITS, ICV, and RRO transmitters.

### 3.3.1. Expected Locations of Event-1 Preseismic Areas

For Event-1, the expected region is found surrounding Graz facility in three cases. One finds that preseismic zones of ITS sunrise anomalies (i.e., the first panel of Figure 3) appears in the eastern part of the Graz station before the Graz sunrise terminator as shown in the first panel of Figure 6. The TBB sunset anomalies (i.e., the third panel of Figure 3) exhibit particular variations, suggesting that the preseismic areas are linked to the Graz sunset terminators and principally to TBB sunset terminators, as displayed in the third panel of Figure 6. The ICV sunset anomalies (i.e., the fourth panel of Figure 3) suggest that the preseismic region occurs mainly close to the Graz sunset terminator and also to ICV sunset terminator, as shown in the fourth panel of Figure 6.

In the western part of the Graz station, a presumed preseismic area, as shown in the second panel of Figure 6, is localized between the ICV and Graz sunrise terminators, taking into consideration the anomaly variations as derived from the second panel of Figure 3. On two occasions, the anomalies appear at the TBB and ICV sunset terminators (i.e., the third and fourth panels of Figure 3, respectively). In this case, the presumed preseismic region is far from the Graz facility, as displayed in the third and fourth panels of Figure 6, respectively, for the TBB and ICV signals. Also, the longest continuous anomaly observations were recorded in ITS and ICV sunrise terminators (i.e., first and second panels of Figure 6) as one can derive from the signal fluctuations shown in the first and second panels of Figure 3. The shortest anomalies correspond to the jumps listed in Table 1, and it concerns the TBB and ICV sunset terminator transmitters, where the jumps occur on 073 DOY according to the third panel and fourth panel of Figure 3. Figure 6 shows in the third and fourth panels, respectively, the preseismic zones for the TBB and ICV jumps.

It comes that the presumed preseismic areas are mainly between the Graz station and the region where the EQs of 22 March 2020 occurred. However, two extensions should also be considered, one to the south-east part up to the TBB transmitter and the other one to the south-west part up to the ICV transmitter.

### 3.3.2. Expected Locations of Event-2 Preseismic Areas

Event-2 shows nearly similar behavior when compared to Event-1. The main presumed preseismic areas are in the vicinity of the Graz station. In the case of the ITS sunset anomalies (i.e., the second panel of Figure 4), one can expect that the preseismic areas are in the eastern part of Graz facility, as displayed in the second panel of Figure 7. It is also the case of RRO sunset anomalies (i.e., the fourth panel of Figure 4), where the preseismic areas appear close to the Graz vicinity in the eastern and western parts, as shown in the fourth panels of Figure 7.

Two far regions from the Graz facility appear in the case of the ITS sunrise and ICV sunset terminators (i.e., the first and third panels of Figure 4. According to the ITS sunrise anomalies, the presumed seismic regions occur in the western part of the Graz station as displayed in the first panel of Figure 7, and for the ICV sunset terminator mainly in the western part, as shown in the third panel of Figure 7. As listed in Table 1, jumps occurred more frequently for Event-2, particularly for RRO transmitters, as shown in the fourth panels of Figure 4. It is interesting to note that the six jumps in RRO sunset anomalies (i.e., the fourth panel of Figure 4) are found before and after the Graz sunset terminators in the eastern and western parts, as displayed in the fourth panel of Figure 7.

Like in Event-1, the presumed preseismic area is in the southern part of Graz facility, with two extensions, one to the south-west part up to the ICV transmitter, and the other to the east part up to the RRO transmitter.

### 3.4. Cross-Correlations of Transmitter Signal Anomalies

We consider hereafter the cross-correlation of the anomalies derived from Figures 3 and 4. The cross-correlation function $P_{xy}(L)$, as given hereafter, enables us to calculate the degree of correlation between two sample parameters X and Y as function of the lag L. It is performed using a routine written in the Interactive Data Language (IDL Version 8.5)

software [32]. We have applied a lag L of 16 days to both EQs. For Event-1, the lag starts on 10 March 2020 (070 DOY) and ends on 25 March 2020 (085 DOY), and for Event-2 from 16 December 2020 (351 DOY) to 31 December 2020 (366 DOY).

$$
P_{xy}(L) \begin{cases} \dfrac{\sum_{k=0}^{N-|L|-1} (x_{k+|L|}-\overline{x}) \ (y_k-\overline{y})}{\sqrt{\left[\sum_{k=0}^{N-1} (x_k-\overline{x})^2\right]\left[\sum_{k=0}^{N-1} (y_k-\overline{y})^2\right]}} & \text{For } L < 0 \\[4ex] \dfrac{\sum_{k=0}^{N-L-1} (x_k-\overline{x}) \ (y_{k+L}-\overline{y})}{\sqrt{\left[\sum_{k=0}^{N-1} (x_k-\overline{x})^2\right]\left[\sum_{k=0}^{N-1} (y_k-\overline{y})^2\right]}} & \text{For } L \geq 0 \end{cases}
$$

The results of the cross-correlations are listed in Table 2, where the EQ events are given in the first column, the sunrise or sunset terminators in the second one, the crossed transmitter signals appear in the third and fourth columns, followed by the columns that indicate the maximum degree of correlation, the corresponding lag, the date (day and month), and finally the day of the year.

**Table 2.** Degree of anomaly correlation derived from transmitter signal anomalies.

| EQ Event | | Transmitter1 | Transmitter2 | Correlation Degree | Lag | Date | DOY |
|---|---|---|---|---|---|---|---|
| Event-1 22 March 2020 | Sunrise | TBB | ICV | 83% | 00 | 10/03 | 070 |
| | | TBB | ITS | 87% | 00 | 10/03 | 070 |
| | | ICV | ITS | 92% | 00 | 10/03 | 070 |
| | | RRO | TBB | 58% | 02 | 12/03 | 072 |
| | | RRO | ICV | 47% | 00 | 10/03 | 070 |
| | | RRO | ITS | 41% | 01 | 11/03 | 071 |
| | Sunset | TBB | ICV | 52% | 00 | 10/03 | 070 |
| | | TBB | ITS | 57% | 04 | 14/03 | 074 |
| | | ICV | ITS | 65% | 04 | 14/03 | 074 |
| | | RRO | TBB | 42% | 01 | 11/03 | 071 |
| | | RRO | ICV | 24% | 03 | 13/03 | 073 |
| | | RRO | ITS | 32% | 04 | 14/03 | 074 |
| Event-2 29 December 2020 | Sunrise | ICV | ITS | 45% | 02 | 18/12 | 353 |
| | | RRO | ICV | 40% | 02 | 18/12 | 353 |
| | | RRO | ITS | 36% | 03 | 19/12 | 354 |
| | Sunset | ICV | ITS | 65% | 05 | 21/12 | 356 |
| | | RRO | ICV | 44% | 03 | 19/12 | 354 |
| | | RRO | ITS | 67% | 02 | 18/12 | 353 |

Event-1 exhibits a high degree of correlation between transmitters, particularly when the lag is equal to zero. Hence, the correlation of TBB and ICV is found to be equal to 83%, with a lag equal to zero corresponding to 070 DOY, i.e., 10 March 2020. The degree of correlation increases to 87% when considering the TBB and ITS signals with a lag equal to zero. The maximum correlation is equal to 92% with a lag also equal to zero and is obtained by the combined ICV and ITS transmitter signals. However, the correlation decreases to less than 60% for the cross-correlations between the RRO and the three other transmitters, as listed in Table 2, with lags of 0, 1, and 2, respectively, when combining RRO-ICV, RRO-ITS, and RRO-TBB. For the sunset anomalies, the degree of correlation is the maximum, equal to 65%, when combining the ICV and ITS signals with a lag equal to 4, i.e., 14 March 2020. It decreases to 57% and 52% when considering, respectively, TBB-ITS and TBB-ICV signals with lags equal to 4 (i.e., 14 March 2020) and to 0 (i.e., 10 March 2020). The correlation is found to be smaller than 45% when combining RRO signals with the three other transmitters. RRO-TBB, RRO-ITS, and RRO-ICV signal correlations exhibit,

respectively, degrees of 42%, 32%, and 24% with lags equal to 1 (i.e., 11 March 2020), 4 (i.e., 14 March 2020), and 3 (i.e., 13 March 2020).

Event-2 reveals a low degree of correlations when compared to Event-1. The TBB transmitter is not used because of a noisy signal occurring in the investigated period. The sunrise correlations are smaller than 50% when combining the ICV, ITS, and RRO signals. A correlation of 45% is obtained when combining the ICV-ITS signals with a lag equal to 2 (i.e., 18 December 2020). A similar lag is found when considering RRO-ICV signals with a degree equal to 40%. The minimum of correlation 36% is attained for the RRO-ITS signals with a lag equal to 3 (i.e., 19 December 2020). The maximum degree of correlation 67% is linked to the RRO-ITS sunset anomalies with a lag equal to 2 (i.e., 18 December 2020), followed by 65% between ICV-ITS signals, and a lag equal to 5 (i.e., 21 December 2020). A correlation equal to 45% appears when combining the RRO-ICV signals with a lag equal to 3 (i.e., 19 December 2020).

It is possible to track the time evolution of the presumed preseismic areas taking into consideration the estimated degree of correlation. For this purpose, we define a correlation scale with different colors, as shown in Figure 8, where the blue is for the degree of correlation higher than 70%, violet smaller than 70% and bigger than 50%, and yellow below 50%. This scale is associated with to the degree of correlation, as listed in Table 2. In addition to the degree of correlation, the corresponding lag is also taken into consideration. In Figure 8 the degree of correlation between two transmitter signal anomalies and the corresponding lag are reported for Event-1 in the six upper panels, three for sunrise correlations and three others for sunset ones, and for Event-2 in the five lower panels, two for sunrise correlations and three for sunset ones.

The three first panels displayed in Figure 8 concern the degree of anomaly correlation associated to the Event-1. The maximum degree of anomaly correlation, more than 80%, for Event-1 is recorded on 10 March 2020 at sunrise and it concerns the ICV-ITS-TBB anomalies. It decreases to 41% when considering the RRO-ITS correlations with a lag on 11 March 2020, and increases again to 58% for the RRO-TBB with a lag on 12 March 2020.

The next three panels refer to Event-1 also, but for sunset anomalies. In this case the degree of anomaly correlation is 42% when combining the RRO-TBB signals with a lag on 11 March 2020. Two days later, the RRO-ICV correlation is found to decrease to 24% with a lag on 13 March 2020, followed by an enhancement of more than 55% with a lag on 14 March 2020 for the ICV-ITS-TBB anomalies. It comes that the region at the southern part of the Graz station exhibits a particularly high anomaly correlation (~80%) in the beginning (i.e., sunrise on 10 March 2020) and decreases (~50%) 4 days later (i.e., sunset on 14 March 2020).

For Event-2, the degree of anomaly correlations is found to be equal to 45% and 36%, respectively, when combining the ICV-ITS and RRO-ITS sunrise anomalies with lags on 18 December 2020 and 19 December 2020, as shown in the seventh and the eighth panels of Figure 8. For the sunset anomalies, displayed in the last three panels of Figure 8, the degree of anomaly correlation is equal to 67% when combining the RRO-ITS signals with a lag on 18 December 2020; it decreases to 44% for the RRO-ICV correlation with a lag on 19 December 2020, and growths again to 65% for the ICV-ITS correlation with a lag on 21 December 2020. It appears that for Event-2, the southern area of Graz station presents a moderate anomaly correlation, smaller than in the case of Event-1.

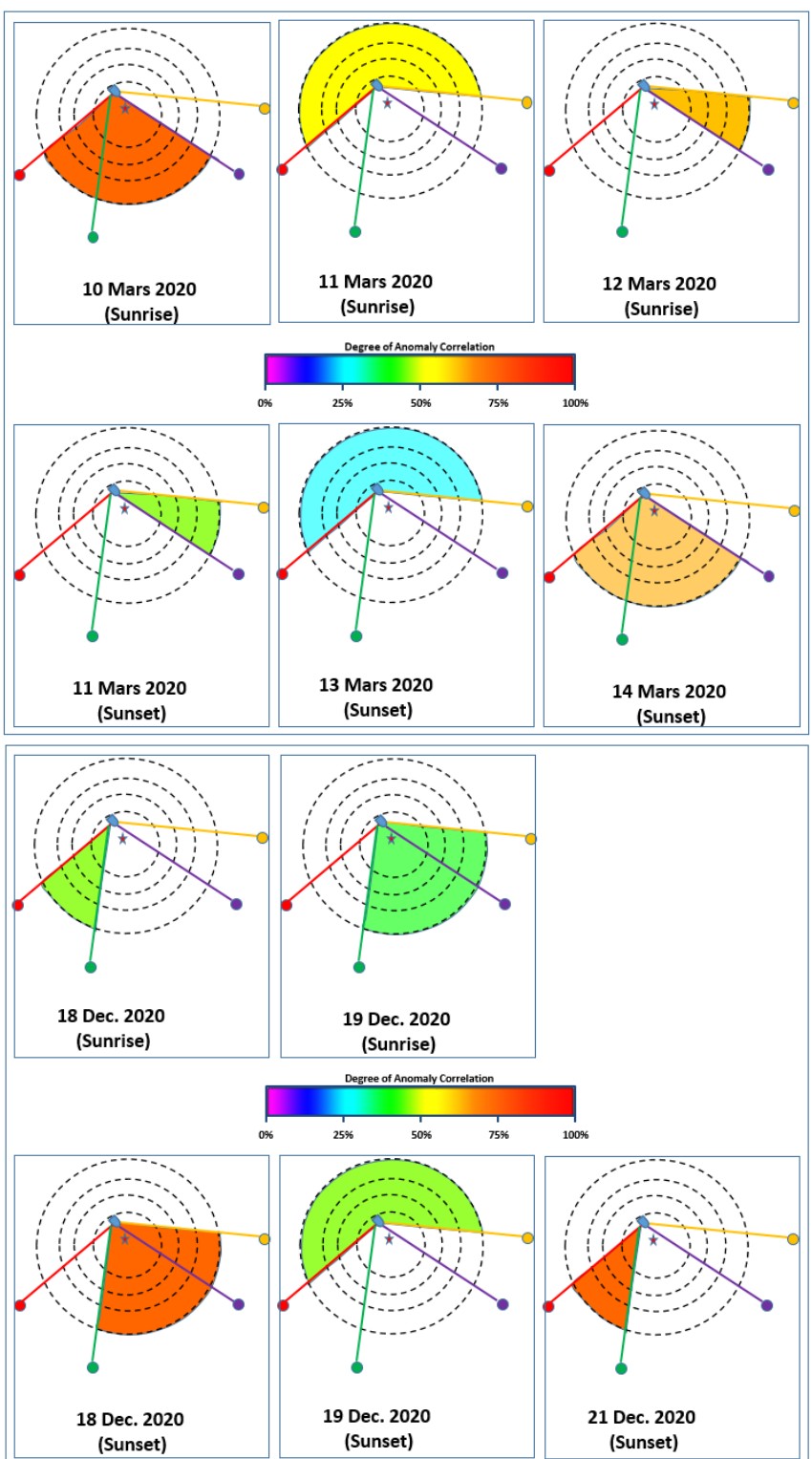

**Figure 8.** Degree of correlation and corresponding lag as derived from Table 2 with a scale where the red and the violet colors correspond, respectively, to high (~90%) and low (~20%) correlations. The red stars correspond to the EQ epicenter. The first six panels indicate the presence of a strong anomaly correlation TBB-ITS-ICV on 10 March 2020 for Event-1 at sunrise. The second five panels refer to the correlation for Event-2. The high correlations appear at sunset between RRO and ITS (18 December 2020), and ICV and ITS (21 December 2020). The blue, red, green, violet, and yellow full small circles designate, respectively, the locations of the Graz station, ITS, ICV, TBB, and RRO transmitters.

## 4. Discussion

The analysis of Croatian EQs was used to characterize the preseismic regions. Three aspects are discussed in this section: the relation between the preseismic areas and Dobrovolsky preparation zone, the core and the eastern and western extensions of the preseismic region, and the role of the cross-correlation to derive when the detection probability of the anomaly is high.

### 4.1. Presumed Preseismic Areas inside Dobrovolsky Preparation Zone

In this work, the investigated regions are defined by the ray path links between the Graz station and the transmitter locations. The analyzed area is limited in the east and west by the longitudes 27.21°E (i.e., TBB transmitter) and 09.71°E (i.e., ICV transmitter), and in the north and south by 47.75°N (i.e., RRO transmitter) and 37.13°N (i.e., ITS transmitter), respectively. This area is bigger than the seismic preparation zone as one can derive from [33] $\rho = 10^{0.43M}$ (km), where $\rho$ is the radius of the preparation zone and M is the EQ magnitude. The radius of the preparation zone is equal to 210 km and 565 km, respectively, for Event-1 ($M_w 5.4$) and Event-2 ($M_w 6.4$). However, both radii are larger than the distances between the EQ epicenters and Graz facility. This can explain the anomalies in the transmitter signals as shown in Figure 6 for Event-1 and Figure 7 for Event-2. Almost all those transmitter terminator anomalies occurred close to the Graz facility. The presumed preseismic areas, from both figures, are inside the Dobrovolsky preparation zone. One can conclude that the area of the ionospheric disturbance at the origin of the transmitter terminator anomaly is comparable with the area derived from [33].

### 4.2. Spatial Dynamic of the Presumed Preseismic Areas

We notice on some occasions that the anomalies happened outside of the preparation zone for Event-1, i.e., the TBB terminator sunset, and for Event-2, i.e., ICV terminator sunset. Additionally, the anomaly hops, as listed in Table 1, should also be considered because of jumps from one terminator to another one as it is the case for Event-1 (see the third and fourth panels of Figure 3 for the TBB and ICV sunsets) and for Event-2 (see the second and third panels of Figure 4 for the ITS and ICV sunsets). It comes out that the ionospheric disturbance behavior is found to be different when compared to those recorded inside the Dobrovolsky preparation zone. In previously cited cases, we note that the shift of the time terminator exhibits a change from one terminator to another in a time interval of about 1 day. For Event-1 (see third panel of Figure 3) the anomaly shift occurs before/after/before the TBB sunset terminators, respectively, on 072 DOY/073 DOY/074 DOY. A similar response happens before/after/before the Graz sunset terminator, respectively, on 072 DOY/073 DOY/075 DOY; this in relation with the ICV transmitter signal (see the fourth panel of Figure 3). In Event-2, two hops (the second panel in Figure 4) are observed for the ITS sunset terminators, respectively, before/after/before on 352 DOY/353 DOY/354 DOY, and on 354 DOY/355 DOY/356 DOY for the Graz terminator. Also, two hops (the third panel in Figure 4) are detected before/after/before ICV sunset terminators on 356 DOY/357 DOY/358 DOY and on 360 DOY/361-362 DOY/363 DOY. This may be interpreted as a particular ionospheric disturbance, occurring on a given day, i.e., 073 DOY for Event-1 and 35 3DOY, 355 DOY, 357 DOY, and 361-362 DOY for Event-2, dragging simultaneously terminator time shifts from the TBB terminators to the Graz terminators (the third panel in Figure 3), and from the Graz terminators to the ICV terminators (the fourth panel of Figure 4). It turns out that inside the preseismic areas, defined by the radius $\rho$ of the preparation zone, anomalies for Event-1 and Event-2 exhibit comparable features despite the difference between $\rho_{Event2}$ and $\rho_{Event1}$ being equal to 355 km. The observed anomalies are mainly in the vicinity of the Graz station. Outside the preparation zone, the anomalies are principally neighboring the transmitter terminators as it is the case for Event-1 with regard to TBB and ICV transmitters, and for Event-2 when considering ICV transmitters. Of course, the trickier situations are the 'jumps' reported in Table 1, where the anomalies are observed inside and outside the Dobrovolsky preparation zone, as shown in the third

and fourth panels of Figure 3 for the TBB and ICV transmitters, and also in the fourth panel of Figure 4 for the ICV transmitters. This may be interpreted as an effect of the dynamics of the preparation zone where the Dobrovolsky radius varies because several days before the earthquake occurrence, the preparation zone is subject to changes due to preseismic energy release dynamics (e.g., longitudinal and latitudinal extensions) also involving ionospheric fluctuations and disturbances at the sunrise and sunset terminators.

*4.3. Anomaly Days Derived from Cross-Correlations of Terminator Time Shifts*

It is evident from the investigated transmitter signals that the anomalous days are those where the terminator time shifts are observed, i.e., those in Figures 3 and 4. However, the cross-correlation of anomaly transmitter signals leads to the enhancement and to the deduction of the more probable anomalous days. The considered lag of about 16 days allows us to find that at least 4 and 5 days, respectively, for Event-1 and Event-2 reveal an important degree of correlation when combining two transmitter terminator anomalies. Hence, in Figures 3 and 4, the anomalies, i.e., the terminator shifts, appear nearly continuously over the 16 investigated days. The use of the cross-correlation permits a reduction of the 16 days to 4–5 days for the terminator anomalies. Also, more probable preseismic areas emerge by considering the ray path between the Graz facility and the radiating transmitters. Hence, it is possible from Figure 8 to define the days and areas where the detection of the anomaly is high. For Event-1 and Event-2, it comes from Figure 8 that the detection anomaly is distinguished on 10 March 2020 (i.e., 12 days before the EQ occurrence) and 18 December 2020 (i.e., 11 days before the EQs). Two areas are linked to these 2 days, one defined per TBB-ITS-ICV transmitters for Event-1 and the other per RRO-ITS transmitters for Event-2. It is not evident to provide a full estimation of the accuracy on the two derived areas for Event-1 at sunrise terminators and Event-2 at sunset ones. However, the applied method, i.e., the estimation of the degree of anomaly correlation, has allowed for the derivation of two regions that are both localized in the southern parts of Graz where it is probable that most of the energy due to the tectonic crust motions is first accumulated in the preparation zone and then released at the rupture point, i.e., the EQ epicenter. The preparation preseismic area 'common' to both events is localized in the southern–eastern part of Graz, also of the Zagreb earthquake epicenters, defined by the geographical triangle TBB-Graz-ITS. This gives indications of the origin of the recorded EQs, which are probably related to the tectonic activities due to the Anatolian plate interactions with African and Arabian plates.

**5. Conclusions**

We have investigated in this study two EQs that occurred in Croatia at distances smaller than 200 km from the Graz VLF/LF facility (Austria). We have used the well-known terminator method where the time shift at sunrise and at sunset of the transmitter signal (i.e., TBB, ICV, ITS and RRO) is combined with the terminators of the reception station (i.e., the Graz facility), as is commonly made [1]. We found that the main time shift anomalies occur inside the Dobrovolsky preparation zone with a radius equal to 210 km and 565 km, respectively, for Event-1 ($M_w$5.4) and Event-2 ($M_w$6.4). This leads us to show that the main core of the preseismic areas are the Graz station and surrounding regions. This means that the area of the ionospheric disturbance at the origin of the transmitter terminator anomalies is of the same order of magnitude as the area derived from [33]. Outside the preparation zone, the anomalies principally neighbor the transmitter terminators, as is the case for the TBB and ICV transmitters for Event-1 and ICV transmitter for Event-2.

The first novelty of our work is the combining of the time shifts of the Graz and transmitter terminators. This has allowed us to discover 'jumps', which mark the 'dynamics' of the anomalies observed inside and outside the Dobrovolsky preparation zone. Such anomaly dynamics are interpreted as changes (i.e., mainly expansions) in the preparation zone where, on two occasions, the preseismic areas were found to develop to the eastern

and western parts of the Graz station. We show for Event-1 that the preseismic region extended up to the TBB transmitter, and for Event-2 to the ICV transmitter.

The second originality of our study is the estimation of the degree of anomaly correlation with a lag of 16 days. This leads us to find that anomalies are confined to 4–5-days instead of 16-days. The degree of correlations was above 80% on 10 March 2020 (i.e., 12 days before the EQ occurrence) for Event-1 and above 65% on 18 December 2020 (i.e., 11 days before the EQ happened) for Event-2. The regions are both localized in the southern parts of Graz and are at the origin of the tectonic accumulated energy that releases at the rupture point, i.e., the EQ epicenter. This may give an indication of the origin of the recorded EQs, which are probably related to the tectonic activities due to the Anatolian plate interactions with African and Arabian plates.

The future perspective will be devoted to using more transmitter signals, at least more than five, which surround the EQs. We insist on the further establishment of a VLF/LF observational network over the world, as it is the case in Europe ([28,34]), in Japan [35], and in India [36]. The cross-correlations of the ray paths may considerably shorten and minimize the preseismic region for the forecasting process. Also, the increase in VLF/LF networks combined with space observations like the China Seismo–Electromagnetic Satellite ([37,38]) can provide ground and spatial coverage of the ionospheric seismic disturbances. The theoretical aspect needs to be considered, specifically the solar terminators effect on the atmospheric gravity waves, as addressed by [18], and the possibility of extracting more physical parameters from the shift of VLF/LF transmitter terminators.

**Author Contributions:** M.Y.B. carried out the study, wrote the initial version of the paper, and took part with H.U.E. and M.S. in the revision of the paper. P.F.B., G.N. and A.E. helped with the data processing of INFREP observations, W.V. and M.P. of the UltraMSK observations. P.H.M.G., M.H. and H.L. took part in the data interpretation. All authors have read and agreed to the published version of the manuscript.

**Funding:** This research received no external funding.

**Data Availability Statement:** The data used in this work are available upon request to the corresponding author. The data are not publicly available due to privacy.

**Acknowledgments:** M.Y.B. and H.E. thank Pier F. Biagi and Masashi Hayakawa who provided help in the installations, here in Graz (Austria), of both reception systems, i.e., INFREP and UltraMSK.

**Conflicts of Interest:** The authors declare no conflicts of interest.

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
