# Peer review of "Analysis of Pre-Seismic Ionospheric Disturbances Prior to 2020 Croatian Earthquakes"

_remotesensing, doi:10.3390/rs16030529_

Round 1
Reviewer 1 Report
Comments and Suggestions for Authors
Dear Authors,
Please find below my general comments on your paper, "Analysis of pre-seismic ionospheric disturbances prior to 2020 Croatian earthquakes":
First of all, I would like to note that your study provides an interesting and original perspective on the pre-earthquake behavior of sub-ionospheric VLF signals.
Suggestions and Questions:
--- Could more details about the terminator method and methodology used in the paper be provided?
--- The conclusions and discussions section could include more emphasis and a summary.
--- More data for future studies and the advantages and potential contributions of using a large network of transmitters could be more comprehensively discussed.
--- There are also some deficiencies and incorrect references in the text. I have attached a PDF file containing specific comments and suggestions for your consideration. I kindly request that you carefully review these and revise your manuscript accordingly. Your attention to these points will undoubtedly elevate the quality and impact of your study.
Thank you for your commitment to enhancing the quality of your work.
Kind regards,

Reviewer 2 Report
Comments and Suggestions for Authors
Your work showcases significant findings, yet refinement could further elevate its impact. Consider refining certain aspects to enhance the study's overall depth and significance. I have provided my suggestions below.
1. Introduction
Line 31: "The earthquake prediction using radio techniques are recognized as efficiency tools" - "are recognized as efficient tools."
Line 34: "Earth’s waveguide where the seismic areas is considered" - "Earth’s waveguide, where the seismic areas are considered"
Line 35: Consider rephrasing the latter part of the sentence for clarity.
Line 49: "Authors estimated the sunrise and sunset terminators times, few days before EQs" - "Authors estimated the times of sunrise and sunset terminators a few days before EQs"
Line 56-58: There seems to be a missing reference or a gap between the sentence mentioning "e.g. for the" and the actual examples of earthquakes.
Line 60-61: "The first explication of the TTs shift has been attributed to the change in the ionospheric electron density, as reported previously." - Consider rephrasing for clarity, e.g., "The initial explanation for the TTs shift attributes it to the change in ionospheric electron density, as previously reported."
Line 67: "Those variations revealed 5-days or 9-11-days periodicities." - Consider rephrasing for clarity, e.g., "These variations revealed periodicities of 5 days or 9-11 days."
Line 71-74: Itis essential to refer some articles that have extensively explored the LAIC mechanism as formulated by Pulinets and Boyarchuk. Some of the recent publications that align well with the context include the following. In my view, these are some of the cited sources that encompass global earthquakes and their reactions to LAIC. Incorporating these references would provide a clearer understanding of LAIC and its potential application.;
https://doi.org/10.3390/atmos14030601
https://doi.org/10.1007/s40328-023-00430-x
https://doi.org/10.1016/j.jseaes.2021.104918
https://doi.org/10.1016/j.asr.2022.04.025
https://doi.org/10.3390/geosciences13110319
Line 84: "A complementary approach has been examined by Pulinets and Boyarchuk (2004)" - Consider rephrasing, e.g., "Pulinets and Boyarchuk (2004) examined a complementary approach..."
Line 89: "Radon is assumed to be the principal source of boundary–layer modifications through air ionization, like the galactic cosmic ray effects on the tropopause." - The sentence structure could be clarified for better understanding.
Line 107: The introduction ends abruptly here, transitioning suddenly to Figure 1. Please ensure that the figure 1 is mentioned in the text.
2. VLF transmitter signal variations before Croatian earthquakes occurrences
The section provides a detailed overview of the earthquakes, the VLF/LF network, and the methodology used for signal analysis. However, there's room for expanding on the methodology steps and relating the observed signal variations more explicitly to seismic activities for a more comprehensive understanding. Additionally, attention to minor typos and consistency in units and coordinates will enhance the clarity of the manuscript.
Line 117, 132: "Crotian" should be corrected to "Croatian."
Line 124: "Decembre" should be corrected to "December."
Line 131: Ensure consistency in writing units - "km" and "UT" (Universal Time) should be kept as abbreviations throughout the text.
Line 151-158: The methodology steps are described but might benefit from more detailed explanations of each step, especially regarding anomaly detection, expected preseismic areas, and the cross-correlation process.
Line 164-171: The explanation of Figure 2 is helpful, but consider expanding on the connection between the signal variations observed in Figure 2 and their relationship with seismic activities to strengthen the analysis.
3. Anomalies as derived from terminator time investigations
314-347: Figures 5, 6, and 7 are mentioned, illustrating the preseismic areas derived from anomalies. These descriptions could be more detailed for a clearer understanding without the visuals.
361-497: The section discusses presumed preseismic areas concerning specific transmitters and anomalies. Consider providing more context and explanations for the observations mentioned in Table 1.
553-627: Figure 8 provides a summary of the degree of correlation between transmitter signal anomalies. The text could further explain the significance of these correlations in understanding preseismic activities.
Overall, while the text describes anomalies, terminator shifts, and correlations between transmitters, it heavily relies on figures. More detailed descriptions and context within the text will enhance the understanding of these anomalies and their correlations without solely depending on visual aids.
4. Discussions
When discussing anomalies occurring outside the preparation zone, ensure a clear comparison with those within the zone. Emphasize differences or similarities to draw precise conclusions about the behavior of these anomalies.
You've highlighted specific temporal patterns in the anomalies, such as shifts occurring in relation to terminator sunset times. It might be useful to delve deeper into potential causes or mechanisms driving these patterns.
The explanation regarding the cross-correlation and its role in identifying probable anomaly days is insightful. However, consider providing a bit more detail about the methodology or statistical significance to support the deduction of these probable anomaly days.
5. Conclusions
Highlight the significance of using the terminator method in determining preseismic areas and the correlation between transmitter anomalies, particularly within the Dobrovolsky preparation zone.
Reinforce the novelty of your approach, emphasizing the innovative aspect of combining transmitter signal shifts with Graz and transmitter terminators. This elucidates the core preseismic areas and their spatial extensions.
Summarize the conclusions drawn regarding the extent of preseismic regions during Event-1 and Event-2, especially their expansions towards TTB and ICV transmitters, respectively.
Comments on the Quality of English LanguageSome minor typos and grammatical errors needs to be well cared of.
Reviewer 3 Report
Comments and Suggestions for Authors
In this manuscript (ms), the authors study the pre-seismic anomalies in the sub-ionospheric Very Low Frequency (VLF) transmitter signals recorded at Graz, Austria, station before the 2020 strong earthquakes (EQs) on 22 March 2020 (Mw5.4) and 29 December 2020 (Mw6.4) at Croatia. They consider the transmitter amplitude variations recorded few weeks before these EQs and successfully identify anomalous variations which are tabulated in Table 1 of page 7. The authors attempt to estimate the location of the EQ preparatory zone by employing a directional analysis presented in Figures 5 to 7. They also present a cross-correlation analysis of the transmitter signal anomalies, which is an innovative part of the ms, that allows a better determination of the anomalous days of transmission that may lead to a better selection of the EQ preparatory areas as discussed in Section 4.3.
The present ms is original and presents a methodology that may be proven very useful for EQ prediction purposes. The subject of the ionospheric disturbances before strong EQs certainly falls within the scope of remote sensing. It is therefore expected that this ms deserves publication in Remote Sensing. The main difficulty, however, is the presentation of the findings that needs to be improved significantly before publication. I list below major and minor points that the authors should consider in their revision:
Major points:
1)The references used should be correctly updated. For example, Reference [4] mentioned in line 47 “They [4] investigated the Kobe earthquake” is irrelevant to Kobe EQ. Instead of [4], the following References are related to the VLF anomalies before Kobe EQ:
Hayakawa, M., Molchanov, O. A., Ondoh, T., and Kawai, E. (1996), The precursory signature effect of the Kobe earthquake on VLF subionospheric signals, J. Comm. Res. Lab., 43(2), 169–180. https://www.nict.go.jp/publication/journal/43/002/Journal_Vol43_No002_pp169-180.pdf
Hayakawa, M., Molchanov, O. A., Ondoh, T., and Kawai, E. (1996), Precursory signature of the Kobe earthquake on VLF subionospheric signal, J. Atmos. Electricity, 16(3) 247-257. https://www.jstage.jst.go.jp/article/jae/16/3/16_247/_pdf
Hayakawa, M., Molchanov, O., Ondoh, T. and Kawai, E., "On the precursory signature of Kobe earthquake on VLF subionospheric signals," 1997 Proceedings of International Symposium on Electromagnetic Compatibility, Beijing, China, 1997, pp. 72-75, https://doi.org/10.1109/ELMAGC.1997.617080 .
Moreover, the ms should adhere to the Remote Sensing style of citations and replace “(Hayakawa, 2015)” in line 699 by some valid Reference from the list of References.
2)The methodology presented in Section 3.3 should be elaborated especially lines 319 to 322, because it was impossible for me to follow how the possible preparatory zones in Figures 6 and 7 are selected. This is very important for the paper.
3)A paragraph discussing the statistical significance of the EQ precursory VLF anomalies should appear somewhere within Section 4.
4)The presentation should be improved significantly by:
4a)filling the gaps that exist in the second column of Table 1 (Please repeat Sunset or any other appropriate symbol for the last 8 lines of this Table).
4b)In lines 555 to 558 there is some discussion about a four-color scale for Figure 8 in which only 3 colors are mentioned which do not correspond to the continuous color palette shown in Figure 8.
Minor points:
There are many typing errors in the ms:
1)l.19 “quakes (EQs)”, l.20 remove the acronym EQ, l.23 “focuses of depth smaller”, l.32 “as effective tools”, l. 35 “areas are considered”, l.61 “first explanation of”, l. 86 “to active”, l.88 “alpha particles formed”, l.111 “in the Section 2.”, l.116 “in Section 4 and summarized in Section 5”, l.303 “from Figure 3 and Figure 4”, l.636 “Eastern”, l.643 “both radii are larger than”, ll.676, 679 “anomalous days” , l.680 “allows to”, l.684 “Also more”, l.690 “to these two days”, l.709 “which permit to”.
2)Figure 1 should be provided with the better quality.
3)The names in the authors contributions should also include their middle names.
In view of the above, the authors should revise their manuscript by addressing the points mentioned above. I would be glad to suggest publication of an appropriately revised ms.
Reviewer 4 Report
Comments and Suggestions for Authors
In the present manuscript, two earthquakes on 22 March 2020 and 29 December 2020 with magnitudes of Mw5.4 and Mw6.4, respectively and epicenters localized close to Zagreb, are examined. An attempt was made to analyze effects in the ionospheric D-region, namely fluctuations before/after the sunrise, sunset, and the cross-correlation of transmitter signals.
General comments
1) Figure 2 does not make it clear when exactly the event occurs. The recommendation is to mark the moment to make it clearer;
2) It is not clear from Figure 3 exactly what effects were produced during Event 1 (EQ on 22 March 2020)?
3) In Figure 4, the bottom two panels showing pre-EQ jumps and treating them as the main effects cannot, in my opinion, be considered EQ indicators because similar jumps are also observed in the pre-EQ period EQ (361 DOY);
4) Line 706-707 the statement has not been proved.
The authors should consider the resulting effects in the D-region, which are most likely of atmospheric origin – planetary waves.
Minor (technical) remarks:
5) Line 712… devoted to use using more transmitter …
6) Line 715 … the possibility to extract of extracting more
7) Line 695 … two EQs that occurred in Croatia …
8) Line 705 At On two occasions …
9) Line 679 allow allows to find …
10) Line 683 Also a more
11) Line 689 one define defined per TBB-ITS-ICV; … linked to this these two days,
12) Line 654 … one as it is the ; … is the cases case for the …;… cases for the Event-1
13) Line 634 … investigated regions is are defined by
14) Line 620 … station exhibits particular particularly high anomaly
15) Line 223 … panel, corresponding correspond to the
16) Line 156 … taking into considerations considerationthe recorded …
17) Line 130 Croatia) were was 7 km …
18) Line 133 … been established since for more than …
19) Line 53 … ionosphere which have has been lowered …
20) Line 48 … transmitter-Inubo-station was of about

Minor editing of English language required
Round 2
Reviewer 4 Report
Comments and Suggestions for Authors
The changes made by the authors have significantly improved the quality of the manuscript.
I propose to the Editorial Board that this paper be accepted in the presented revised version.
Author Response
We would like to thank the referee for taking the time to review our manuscript